# The Impact of New infrastructure Construction on Optimization and Upgrading of Industrial Structure

**Wei Wu** [1,2], **Zheng Ji** [3,\*] and **Han Liang** [4]

1   Business School, Jiangsu University of Science and Technology, Zhangjiagang 215600, China;
    wuwei@just.edu.cn
2   Yangtze River Delta Social Development Research Center, Jiangsu University of Science and Technology,
    Zhangjiagang 215600, China
3   National School of Development and Policy, Southeast University, Nanjing 211189, China
4   Dong Fureng Economic and Social Development School, Wuhan University, Wuhan 430072, China;
    2021106270016@whu.edu.cn
\*   Correspondence: jz0429@163.com

**Abstract:** Industrial optimization is needed as China's economy moves toward high-quality growth. The construction of new infrastructure, driven by new development concepts and patterns, facilitates industrial optimization. This study aims to explore the impact and mechanisms of new infrastructure construction on industrial structure optimization. The index of industrial structure upgrading and the pace of industrial transformation were calculated using panel data from 266 prefecture-level cities, which spanned from 2011 to 2018. This study uses two-way fixed effects and mediation effect methodologies to experimentally investigate the impact of new infrastructure construction on the optimization of industrial structure, while also considering possible endogeneity concerns. We found that new infrastructure building promotes industrial transformation and urban industrial structure upgrading. These results pass robustness and endogeneity testing. However, the impact of new infrastructure construction on industrial structure upgrading varies across cities. There is a significant driving effect in economically larger cities with better traditional infrastructure, and those in the eastern region. Smaller cities and those with inferior infrastructure have less impact. New infrastructure construction optimizes urban industrial structures through technological innovation and professional agglomeration, according to mediation effect study. Diversified agglomeration does not significantly upgrade industrial structures. The limitations of our model include the fact that the data does not describe industrial structural dynamics and it does not apply on other geographic scales. We illuminate the intricate connection between new infrastructure and industry upgrading by including city heterogeneity and the mediating impacts of technical innovation and professional clustering.

**Keywords:** new infrastructure; industrial structure upgrading; speed of industrial transformation; technological innovation; industrial agglomeration

## 1. Introduction

China's economy has entered a new stage with high-quality development as the main theme, but it still faces the dilemma of intertwined structural, systemic, and cyclical problems. Breaking the cycle of obstacles in industrial structure and continuous structural adjustment, particularly the adjustment of industrial structure, is of the utmost importance [1]. The construction of new infrastructures, guided by new development concepts and patterns, has become a crucial approach to economic transition and structural adjustment. New infrastructure construction will provide the infrastructure for a new round of technological revolution and industrial transformation, with the digitization of information technology at its core, and it is a significant cornerstone for the development

of the digital economy [2]. This includes not only information infrastructure construction but also digital transformation of traditional infrastructures [3]. New infrastructure construction enhances the self-innovation ability of upstream and downstream industrial chains [4], breaks through spatio-temporal barriers [5], reduces transaction costs, increases the cooperativity between supply and demand [6], and provides innovative conditions and enabling means for fundamental changes in economic and social development. Therefore, can "new infrastructure construction" facilitate the upgrading of industrial structure? What are the influencing mechanisms? Are there any heterogeneous situations?

Since the proposal of new infrastructure construction in 2018, efforts to accelerate its progress have been made from central to local governments, and academic research on it is increasingly rich. Based on the definition of new infrastructure by the National Development and Reform Commission, different scholars have measured one aspect of information infrastructure [7], integrated infrastructure [8], and innovative infrastructure [9]. Guo et al. (2020) [10] believe that new infrastructure construction needs to be rooted in generation-breaking technology and is of great significance for promoting the transformation of old and new drivers, upgrading traditional manufacturing industries, and achieving inclusive economic growth. At the same time, new infrastructure construction is an infrastructure building initiative driven by technology, which can effectively promote the integrated development of the digital economy and industries through the implementation of bottom-up technological promotion, innovative platform mobilization, and industry integration, which can accelerate the transformation of intelligent manufacturing [11]. Some scholars have studied the mechanisms of artificial intelligence on economic structural transformation [12], industrial transformation [13], high-quality economic development [14], the paths of new infrastructure construction to empower high-quality economic development [15], and the transformation and upgrading of manufacturing [16]. Others have studied the positive impact of network construction on economic growth [17], innovative development [18], and rural development [19].

The theoretical background of the relationship between new infrastructure and the optimization and upgrading of the industrial structure lies in the concept of the digital economy and its impact on economic development. The digital economy refers to the economic activities that are based on digital technologies, such as the internet, artificial intelligence, and big data. It has been recognized as a key driver of growth, innovation, and productivity in the modern era [6–8]. There are several ways in which building new infrastructure provides a new impetus to optimize the industrial structure. New infrastructure—including high-speed internet, cloud computing, and data centers—enables businesses to adopt digital technologies and processes, leading to increased efficiency, innovation, and productivity [5]. The relationship between new infrastructure and industrial structure upgrading is not sufficiently systematic and in-depth, and the effect of industrial structure upgrading by new infrastructure has not been empirically tested. Analyzing this issue helps to clarify the effect of "new infrastructure" on industrial structure upgrading. This paper, based on urban panel data and using benchmark regression, mediation effect testing, and other methods, examines the impact of new infrastructure on the optimization and upgrading of the industrial structure from the perspectives of heterogeneity and impact mechanisms, and uses instrumental variables to solve endogeneity problems.

This paper's marginal contribution lies in several factors. First, this paper calculates the index of urban industrial structure upgrading and industrial transformation speed, based on the dynamic changes between industries, and expands the analysis of the influencing factors of industrial structure optimization. This is different from the previous use of provincial level data to construct industrial structure indicators. Second, through heterogeneous analysis, this paper measures the effects of new infrastructure on optimizing and upgrading the industrial structure of cities with different locations, different economic scales, and different levels of traditional infrastructure development. The results show that, under different constraints, the effect of urban industrial structure upgrading and optimization varies, and the effect of urban industrial structure upgrading with new infrastructure

is not always positive and significant at the current stage. Third, from the perspective of the two dimensions of technological innovation and industrial agglomeration, this paper explores the "black box of transmission path" of new infrastructure in optimizing the industrial structure, i.e., how new funds, through promoting technological entrepreneurship and promoting "industrial geographical agglomeration" to "industrial interconnection" and other paths, have accelerated the speed of industrial transformation and improved the standard of industrial structure.

## 2. Materials and Methods

### 2.1. New Infrastructure and Industrial Structure Optimization and Upgrading

Infrastructure development helps to promote economic growth and technological improvement, and it can also have a significant impact on industrial structure upgrading [20]. The impact of new infrastructure on industrial upgrading can be reflected both in the degree of integration and maturity embedded in the modern industrial system, as well as empowered directions and features for high-quality economic development [21]. The mechanism of new infrastructure in promoting industrial upgrading is similar to traditional infrastructure. As the leading capital, infrastructure has significant positive externality and multiplier effects [22], which can increase employment [23], improve the business environment [24], and is an important driver of economic growth [25], providing assurance for industrial upgrading. Infrastructure optimizes resource allocation [26], especially with regard to the construction of major infrastructure; can realize the concentration and cross-regional flow of resources [27]; reduce the intermediate costs of enterprises and expand their geographic range of development; and can optimize the structure of production and organization, thus providing environmental conditions for industrial structure optimization and promoting industrial agglomeration and structural adjustment [28]. Infrastructure promotes the construction of market integration [29]; to some extent breaks spatio-temporal barriers, expands the spillover range of knowledge technology and other resource elements; enhances market accessibility and transaction rate [30]; promotes the agglomeration of advantageous resource elements [31]; and drives industrial upgrading. Infrastructure construction will generate network linkage effects [32]; transportation and communication infrastructure in particular can promote the play of industrial network linkage effects [33], which is conducive to knowledge spillover and cooperation between enterprises [34], thus promoting industrial upgrades.

Based on the analysis, this paper proposes H1:

**Hypothesis 1 (H1).** *New infrastructure can promote the optimization and upgrading of the industrial structure.*

### 2.2. The Impact Mechanism of New Infrastructure on Industrial Structure Optimization and Upgrading

#### 2.2.1. Technological Innovation Effect

Technological innovation can strongly promote the upgrading of industrial structure. The new infrastructure construction is likely to bring about industry siphoning effects, promoting industry agglomeration and enhancing industry technological innovation efficiency, and will hence play a significant role in industrial structure upgrading. On the one hand, new infrastructure can enhance the innovative capacity of industry and promote the upgrading of industrial structure with the help of the diffusion effect of technological spillover. For instance, the technological knowledge included in new digital infrastructures like big data and artificial intelligence belongs to general technology. Using the effect of technological spillover, the integrated application of AI with cloud computing and big data can bring innovative ideas to other fields [35]. Strengthening the construction of new infrastructure can help to realize innovative changes in digital production, networked business models, and intelligent management paradigms, thereby improving the technological innovation capability and efficiency of enterprises. On the other hand, the development

of digital technologies like big data, cloud computing, and artificial intelligence requires a large amount of data support. The open AI material database has provided great convenience for AI theoretical research and application algorithm development, with many innovative activities not needing to start from scratch, thereby significantly improving the efficiency and success rate of R&D and innovation activities [36].

After the analysis, this paper proposes H2:

**Hypothesis 2 (H2).** *New infrastructure can promote the upgrading of the industrial structure by facilitating technological innovation.*

### 2.2.2. Industrial Agglomeration Effect

According to the new geographical economic theory, upstream and downstream associated enterprises tend to be concentrated due to the transportation cost considerations under the influence of economies of scale [37]. The agglomeration and production in the same region intensify enterprise competition, coupled with the increasing demand for specialized services and refined division of labor in the market, which leads to continuously rising costs for manufacturing enterprises seeking personalized design and R&D breakthroughs. Therefore, enterprises choose to outsource some production links to specialized productive service companies, ultimately forming industrial clusters of productive services based around manufacturing [38]. This agglomeration model effectively utilizes the advantages of economies of scale in the production of intermediate services and products, thereby promoting the transformation of production links toward high added value [39,40]. To a certain extent, new infrastructure breaks through spatio-temporal barriers, promotes the information dissemination between productive service agglomeration and manufacturing enterprises [41], aids the connection between upstream and downstream of the industrial chain, and thus propels the industry from "geographical agglomeration" towards a deeper level of "industrial interconnection". The development of new infrastructure also accelerates market development and technological diffusion in productive services within the agglomeration area, thereby promoting industrial upgrading.

Based on the analysis, this paper proposes H3:

**Hypothesis 3 (H3).** *New infrastructure promotes the upgrading of the industrial structure through facilitating industrial agglomeration.*

### 2.3. Heterogeneity of New Infrastructure Impact on Industrial Structure Optimization and Upgrading

Although China has entered the stage of high-quality development, the problem of unbalanced regional development still exists, with different cities showing significant differences in resource endowment. Compared with eastern cities, cities in central and western China have significant gaps in resource aggregation and technological development. The degree of resource aggregation and the level of human capital are relatively low, resulting in differential impacts of new infrastructure on industrial structure upgrading. At the same time, in cities in central and western regions, the level of digitization, networking, and intelligentization of traditional infrastructure is relatively low, and the level of new infrastructure development is not high, which may affect the incentive role of new infrastructure in promoting industrial structure upgrading. Based on the analysis, this paper proposes the following hypotheses:

**Hypothesis 4 (H4).** *The effect of new infrastructure in promoting the upgrading of the industrial structure in developed cities is stronger than in less developed cities.*

**Hypothesis 5 (H5).** *The incentive role of new infrastructure in promoting industrial structure upgrading is stronger in cities with better traditional infrastructure.*

*2.4. The Model*

This article constructs a city linear panel benchmark test model to explore the impact of new infrastructure on industrial structure upgrading. The model is as follows:

$$industry_{i,t} = \alpha_0 + \alpha_1 NI_{i,t} + \alpha_2 control_{i,t} + \phi_i + \lambda_t + \varepsilon_{i,t}, \tag{1}$$

where $i$ denotes the city and $t$ denotes time. *industry* stands for the index of industrial structure optimization, which is the dependent variable in this paper. $NI$ represents the level of new infrastructure development, which is the core independent variable in this paper. To control for omitted variable bias, a series of control variables affecting industrial structure optimization are selected, represented by *control*. $\phi$ stands for unobservable area factors that do not vary with time, to control for area fixed effects. $\lambda$ stands for unobservable factors that only vary with time but not with individuals, to control for the fixed effects of time. $\varepsilon$ represents the independently and identically distributed random error term.

*2.5. Data and Variables*

This paper uses panel data of 266 prefecture-level cities in 29 provinces in China from 2011 to 2018 to assess the impact of new infrastructure on the optimization of industrial structure. The research data sources include the "China City Statistical Yearbook", EPS data platform, China Research Data Service Platform (CNRDS), etc., and statistical data are matched according to the unique identification code of the cities. The specific variables involved in this paper are as follows:

(1)  Dependent variable: Industrial structure optimization index (industry)

Existing research often reflects the optimization of the industrial structure through changes in the proportion of GDP of the three industries [42,43]. The change in the GDP ratio is an important dimension of industrial structure optimization, but the optimization of the industrial structure should also include improvements in labor productivity. According to Clark's theorem, industrial structure optimization is defined as the increase in the proportion of non-agricultural industries, and measurement indicators, such as the proportion of high-tech industries and Moore's index, can be used to measure the sophistication of the industrial structure. However, these indicators measure the dynamic evolution of the industrial structure from a quantitative perspective without capturing the intrinsic nature of industrial structure upgrading, and could easily lead to overestimated values. Therefore, the optimization of industrial structure should not only reflect an increase in quantity, but also an improvement in quality. To fully reflect the connotations of the industrial structure, this paper measures two dimensions: the sophistication of industrial structure and the speed of industrial transformation.

Firstly, following the research of Liu [44], an advanced industrial structure index (AIS) is constructed to depict the relationship between the ratio of output value among industries and labor productivity. The calculation process is as follows:

$$AIS_{(i,t)} = \sum_{a=1}^{3} PI_{i,a,t} \times LP_{i,a,t}, \tag{2}$$

where $PI_{i,a,t}$ denotes the share of value added of industry $a$ in city $i$ in period $t$ in the city's GDP. $LP_{i,a,t}$ denotes the labor productivity of industry $a$ in city $i$ at time $t$, measured by the ratio of added value to employment in that industry:

$$LP_{i,a,t} = Pi_{i,a,t} / PN_{i,a,t}, \tag{3}$$

where $Pi_{i,a,t}$ denotes the added value of industry $a$ in city $i$ during time period $t$. $PN_{i,a,t}$ denotes the number of employees engaged in industry $a$ in city $i$ during time period $t$. $Pi_{i,a,t}$ is dimensionless, and $PN_{i,a,t}$ is quantitative. In this paper, $PN_{i,a,t}$ is normalized in this paper to eliminate the effect of the quantitative scale.

In addition, referencing the research of Yuan (2018) [45], this paper constructs an index of the speed of industrial transformation (SIT), reflecting the dynamic evolution process from a low level to a high level based on the quantity level of the three major industries according to the sequence of economic development. The calculation formula is as follows:

$$SIT_{i,t} = \sum_{a=1}^{3} PI_{i,a,t} \times a. \tag{4}$$

This index can reflect the evolutionary trend of industrial structure from the primary industry to the secondary and tertiary industries. The larger the SIT value, the more optimized the industrial structure.

(2) Core Independent Variable (New Infrastructure, NI)

New infrastructure is based on informatization and focuses on technological advancements. It is closely related to the development levels of artificial intelligence, 5G, the Internet of Things (IoT), and the industrial internet. In other words, the development of new infrastructure in a region is highly correlated with the level of information infrastructure construction [1]. Therefore, this paper uses the development level of information infrastructure to reflect the status of new infrastructure in a city. Scholars have used indicators such as broadcasting and telephone service prices, telephone penetration rates, fiber optic cable length, and total volume of postal and telecommunications services to measure information infrastructure development [46,47]. Additionally, the "Broadband China" strategic pilot city indicators from the Ministry of Industry and Information Technology. To this end, this study uses Principal Component Analysis (PCA) to objectively weight the indicators, taking into account regional and temporal factors, extending the connotation of information infrastructure indicators. This study estimates the level of information infrastructure development in each city by using four indicators: internet penetration rate, relevant practitioners, relevant output, and mobile phone penetration rate [48]. The specific corresponding indicators are the number of internet users per hundred people, the proportion of computer services and software practitioners to urban employment, per capita telecommunications service volume, and the number of mobile phone users per hundred people. Principal Component Analysis (PCA) is also used to standardize and reduce the dimensionality of the above indicators, ultimately obtaining a comprehensive index of information infrastructure as a proxy variable for new infrastructure construction.

(3) Control Variables

To mitigate endogeneity bias caused by omitted variables, other factors that have an impact on regional industrial structure upgrading were selected as control variables, drawing from existing research findings. Population density (pop) is characterized by the ratio between the year-end total population (in thousands) and the administrative area land area (in square kilometers). The level of human capital (edu) is measured by the ratio between the number of regular undergraduate and postgraduate students (in thousands) and the year-end total population (in thousands). Foreign direct investment (fdi) is measured by the proportion of actual utilization of foreign investment (in thousands of US dollars) to the regional GDP (in thousands of RMB). The actual utilization of foreign investment is converted at the middle rate of the Chinese yuan to US dollar exchange rate for that year. Fixed asset investment (invest) is represented by the ratio of the total amount of fixed asset investment in the city (in thousands of RMB) to the regional GDP (in thousands of RMB). The level of unemployment (unemp) is indicated by the ratio between the number of urban registered unemployed people (in thousands) and the year-end total population (in thousands).

Table 1 reports the basic characteristics of the main variables. The mean of industrial structure upgrading is 6.470, with a standard deviation of 0.350. The maximum value is 7.610, and the minimum value is 5.520. This indicates that there are differences in the level of industrial structure upgrading among the sample cities, with significant variations

during the observation period. The mean of industrial transformation speed is 2.270, with a standard deviation of 0.140. The maximum value is 2.800 and the minimum value is 1.830. This indicates that the industrial transformation speed during the observation period varies across sample cities, but with a relatively small difference. This also suggests that the speed of industrial transformation is approaching among different cities in China. From the data, there is a significant gap between industrial structure upgrading and industrial transformation speed. Choosing either indicator alone cannot accurately reflect the true level of urban industrial structure optimization. This highlights the necessity of characterizing industrial structure optimization from both dimensions, i.e., industrial structure upgrading and industrial transformation speed. The core independent variable, new infrastructure development, and other control variables also show variations among the sample cities, providing empirical material for investigating the impact of new infrastructure development on the optimization and upgrading of industrial structure.

**Table 1.** Descriptive statistics for main variables.

| Variables | Mean | SD | p50 | Min | Max |
|-----------|------|------|--------|--------|-------|
| AIS | 6.470 | 0.350 | 6.440 | 5.520 | 7.610 |
| SIT | 2.270 | 0.140 | 2.260 | 1.830 | 2.800 |
| NI | 0.030 | 1.010 | −0.220 | −1.540 | 14.62 |
| pop | 5.740 | 0.960 | 5.880 | 1.630 | 9.980 |
| edu | 0.050 | 0.090 | 0.040 | 0 | 2.620 |
| fid | 0.050 | 0.050 | 0.030 | 0 | 0.730 |
| invest | 0.790 | 0.340 | 0.750 | 0 | 5.600 |
| unemp | −0.440 | 1.630 | −0.460 | −4.610 | 4.010 |

## 3. Results

### 3.1. Baseline Regression

To control for regional macroeconomic variations and differences between regions that do not change over time, this study employs a fixed-effects model as the benchmark regression. Industrial structure, as a critical measure of economic development quality, directly affects the level of economic development through its overall growth and structural adjustments. Table 2 presents the regression results of the impact of new infrastructure construction on the upgrading of industrial structure and the transformation speed of industries. The models show relatively high goodness-of-fit, and the regression coefficients are robust with standard errors. In column (1), only the upgrading of urban industrial structure (AIS) is used as the dependent variable without controlling for other variables. The results indicate that new infrastructure construction significantly promotes the optimization of industrial structure. In column (2), control variables are added to column (1). The results show that the coefficient of new infrastructure construction is significantly positive at the 1% level, suggesting that it significantly promotes the upgrading of industrial structure. In column (3), only the transformation speed of urban industries (SIT) is used as the dependent variable. The results indicate that new infrastructure construction significantly accelerates industrial transformation. Column (4) presents the regression results of the transformation speed of urban industries (SIT) with control variables added. It can be observed that new infrastructure construction still significantly enhances the transformation speed of industries. Overall, the estimated coefficients of new infrastructure construction on the upgrading of industrial structure (AIS) and the transformation speed of industries (SIT) are significantly positive at a level higher than 5%. This indicates that new infrastructure construction plays a strong positive incentivizing role in optimizing industrial structure. In other words, as the level of new infrastructure construction development increases, it provides favorable conditions for driving industrial upgrading and optimization. This validates hypothesis H1.

**Table 2.** Impact of new infrastructure on industrial structure: baseline regression.

| Variables | AIS | | SIT | |
|---|---|---|---|---|
| | **(1)** | **(2)** | **(3)** | **(4)** |
| NI | 0.0064 *** | 0.0052 *** | 0.0028 ** | 0.0023 ** |
| | (0.0024) | (0.0018) | (0.0012) | (0.0010) |
| pop | | 0.0142 *** | | 0.0069 ** |
| | | (0.0050) | | (0.0027) |
| edu | | 0.0373 *** | | 0.0179 *** |
| | | (0.0091) | | (0.0038) |
| fdi | | −0.0765 | | −0.0275 |
| | | (0.0553) | | (0.0302) |
| invest | | −0.0041 | | −0.0026 |
| | | (0.0080) | | (0.0038) |
| unemp | | −0.0053 | | −0.0021 |
| | | (0.0036) | | (0.0017) |
| Constant | 6.4661 *** | 6.3661 *** | 2.2695 *** | 2.2218 *** |
| | (0.0001) | (0.0300) | (0.0000) | (0.0161) |
| Observations | 1876 | 1414 | 1876 | 1414 |
| Adjusted R-squared | 0.9767 | 0.9878 | 0.9675 | 0.9823 |
| year FE | YES | YES | YES | YES |
| city FE | YES | YES | YES | YES |
| F | 7.276 | 4.965 | 4.959 | 5.225 |

The standard errors are shown in parentheses. ***, **, and * represent significance at the 1%, 5%, and 10% levels, respectively.

The effectiveness of new infrastructure construction in optimizing the industrial structure varies. It was found that the estimated coefficient for the upgrading of industrial structure is 0.0064, while the estimated coefficient for the transformation speed of industrial structure is 0.0023. This is significantly lower than the coefficient for the upgrading of industrial structure. It suggests that new infrastructure construction has a greater potential to optimize the upgrading of industrial structure compared to its impact on the transformation speed of industries.

### 3.2. Robustness

To enhance the empirical basis and ensure robustness in the regression results, this study further conducts additional tests by incorporating time trend components, considering outliers in the observations, and employing instrumental variable techniques.

3.2.1. Time Trends Based on Latitude

The shift in industrial structure may be influenced by regional spatial disparities and time trends. Although the benchmark model already controls for city and time fixed effects, exploring the interactive effects between the two variables on the industrial structure optimization effect of the new infrastructure construction is still worthwhile. Therefore, this study incorporates time trend components based on latitude (the interaction term between city latitude and year, latitude*year) into the baseline model.

The results in Table 3, columns (1) and (2), show that the estimated coefficients for the impact of new infrastructure construction on industrial structure upgrading and transformation speed are 0.0055 and 0.0025, respectively, both significant at the 1% level. This indicates that the industrial structure optimization effect of new infrastructure construction is robust, and its effect on industrial structure upgrading is stronger than its effect on transformation speed. Additionally, the positive coefficient of the interaction term latitude*year reflects the spatial characteristics of regional industrial structure optimization, indicating that regions with higher latitudes exhibit a greater industrial optimization effect of new infrastructure construction. This to some extent reveals the "North-South development gap" in China's economic development quality. The reason may lie in the

fact that regions with lower latitudes mainly include the Yangtze River Delta, the Pearl River Delta, and the Guangdong-Hong Kong-Macao Greater Bay Area—these regions have undergone early reform, achieved rapid economic development, and attained high levels of technological innovation. Therefore, their industrial transformation preceded that of regions with higher latitudes. Comparatively, regions with lower latitudes have limited absorption of the industrial structure optimization effect of new infrastructure construction, whereas the absorption effect is significant in regions with higher latitudes.

**Table 3.** Robustness.

| Variables | Time Trends of Latitude Controlled | | Outliers | |
|---|---|---|---|---|
| | **(1)** | **(2)** | **(3)** | **(4)** |
| NI | 0.0055 *** | 0.0025 *** | 0.0072 * | 0.0022 ** |
| | (0.0017) | (0.0009) | (0.0044) | (0.0009) |
| latitude | 0.0004 ** | 0.0003 *** | | |
| | (0.0002) | (0.0001) | | |
| pop | 0.0138 *** | 0.0066 ** | 0.0091 | 0.0091 ** |
| | (0.0050) | (0.0027) | (0.0072) | (0.0038) |
| edu | 0.0359 *** | 0.0170 *** | 0.1446 | 0.1094 ** |
| | (0.0084) | (0.0035) | (0.0909) | (0.0444) |
| fdi | −0.0959 * | −0.0400 | −0.1489 ** | −0.0370 |
| | (0.0523) | (0.0279) | (0.0722) | (0.0371) |
| invest | −0.0010 | −0.0006 | 0.0053 | −0.0048 |
| | (0.0070) | (0.0032) | (0.0132) | (0.0059) |
| unemp | −0.0060 * | −0.0026 | −0.0075 ** | −0.0022 |
| | (0.0034) | (0.0016) | (0.0036) | (0.0017) |
| Constant | −22.5312 ** | −16.3297 ** | 6.3864 *** | 2.2062 *** |
| | (11.3956) | (6.2969) | (0.0443) | (0.0232) |
| Observations | 1414 | 1414 | 1414 | 1414 |
| Adjusted R-squared | 0.9881 | 0.9829 | 0.9846 | 0.9824 |
| year FE | YES | YES | YES | YES |
| city FE | YES | YES | YES | YES |
| F | 6.089 | 6.711 | 2.268 | 3.358 |

The standard errors are shown in parentheses. ***, **, and * represent significance at the 1%, 5%, and 10% levels, respectively.

### 3.2.2. Outliers

To eliminate outliers and mitigate interference from data fluctuations, we winsorize the dependent variable, independent variables, and control variables, with upper and lower 1% tails winsorized. The results in Table 3, column (3), indicate that after winsorization, the coefficient of the core independent variable, new infrastructure construction, is 0.0072, which is statistically significant at the 10% level. Moreover, in Table 3, column (4), the coefficient of the core independent variable, new infrastructure construction, is 0.0022, which is statistically significant at the 5% level. This suggests that the industrial optimization effect of new infrastructure construction remains robust, as it continues to exert a greater influence on upgrading the industrial structure compared to enhancing the speed of industrial transformation. These findings are consistent with the baseline regression results mentioned earlier.

### 3.2.3. Tests for Instrumental Variables

Table 4 utilizes the number of mobile phone users as an instrumental variable to re-estimate the benchmark model using the two-stage least squares (2SLS) method. The selection of mobile phone users as the instrumental variable satisfies the requirement of a correlation between the instrumental variable and the endogenous independent variable, namely, the relationship with new infrastructure development. Additionally, it meets the re-

quirement of irrelevance between the instrumental variable and the disturbance term. New infrastructure construction in China is a government policy that is systematically promoted from top to bottom. As of 2022, the number of mobile phone users in China has reached 1.683 billion. The 19th National Congress of the Communist Party of China also explicitly stated the objective of accelerating the development of the digital economy, promoting the deep integration of the digital and real economies, and creating internationally competitive digital industry clusters. Optimizing infrastructure layout, structure, functionality, and system integration is crucial for establishing a modern infrastructure system. Therefore, the number of mobile phone users plays a role in the digital economy wave because of national policy impact and government promotion, exhibiting exogeneity.

**Table 4.** Tests for IVs.

| Variables | NI | AIS | SIT |
|---|---|---|---|
| | (1) | (2) | (3) |
| IV | −0.5150 ** | | |
| | (0.2422) | | |
| NI | | 1.3488 * | 0.5519 * |
| | | (0.7458) | (0.3059) |
| pop | −0.1700 | −0.1321 | −0.0541 |
| | (0.1116) | (0.1232) | (0.0506) |
| edu | −0.4445 *** | −1.2758 | −0.5036 |
| | (0.1181) | (1.0751) | (0.4411) |
| fdi | 0.5762 | −1.6838 | −0.7257 |
| | (0.4507) | (1.1038) | (0.4528) |
| invest | 0.0223 | 0.1891 | 0.0864 |
| | (0.0452) | (0.2216) | (0.0909) |
| unemp | −0.0011 | 0.1614 | 0.0717 |
| | (0.0285) | (0.1404) | (0.0576) |
| Constant | 4.9782 ** | | |
| | (2.1484) | | |
| Observations | 1408 | 1387 | 1387 |
| Adjusted R-squared | 0.7075 | −14.1479 | −15.5391 |
| year FE | YES | YES | YES |
| city FE | YES | YES | YES |
| F | 3.612 | 6.556 | 5.569 |

The standard errors are shown in parentheses. ***, **, and * represent significance at the 1%, 5%, and 10% levels, respectively.

Table 4 presents the results of the two-stage least squares (2SLS) regression. In the first column, the instrumental variable (IV) estimate coefficient from the first-stage regression is −0.5150, and it is significant at the 5% level. Columns (2) and (3) of Table 4 present the results of the second-stage 2SLS regression, showing the marginal impact coefficients of new infrastructure on industrial structure optimization, which are 1.3488 and 0.5519, respectively. Both coefficients are significant at the 10% level, indicating that new infrastructure plays an important role in the process of industries transitioning to higher-end sectors. In addition, the rationality of selecting the number of mobile phone users as an instrumental variable is further validated by the instrumental variable tests. The results of the instrumental variable tests indicate that the model does not suffer from the problems of unidentifiability and weak instrumental variables. The Anderson canon. corr. LM statistic value is 19.705, with a *p*-value of 0.0000, rejecting the null hypothesis of unidentifiability. The Cragg–Donald Wald F statistic value is 19.440, which is greater than the empirical threshold of 10, passing the weak instrumental variable test. Therefore, it can be observed that after alleviating the potential endogeneity issue, the regression results using instrumental variables are consistent with the previous conclusions, demonstrating that the selected instrumental variables are robust and confirming the hypothesis proposed in this study.

*3.3. The Heterogeneity Test of New Infrastructure on Industrial Structure Optimization*

To avoid potential omitted variable bias in the baseline conclusion and provide evidence for a better understanding of the boundary conditions for digital finance-driven industrial structure optimization, the following analysis examines the differential effects of digital finance on industrial structure optimization from aspects such as urban location, economic scale, and traditional infrastructure construction.

### 3.3.1. Location Heterogeneity

Given the significant spatial disparities in China, location conditions may be one of the factors that affect the effectiveness of new infrastructure in optimizing industrial structure. Consequently, the sample cities are divided into two groups: eastern cities and central-western cities, in order to further validate the heterogeneous impact of spatial location on industrial structure.

The results of heterogeneity tests in Table 5 show that the regression coefficients of new infrastructure are positive for both eastern and central-western cities, indicating a positive motivating effect of new infrastructure construction on the industrial structure optimization of any given city. However, the impact of new infrastructure varies depending on location, with a more pronounced stimulating effect observed in central-western cities. This suggests that new infrastructure is particularly beneficial for optimizing the industrial structure in central-western cities. Furthermore, even after considering urban heterogeneity, the driving effect of new infrastructure on the advancement of industrial structure is still greater than its effect on accelerating the pace of industrial transformation, which is aligned with previous research findings. The impact of new infrastructure on industrial structure optimization is significantly stronger in central-western cities compared to eastern cities. This phenomenon could be attributed to the relatively stable industrial structure in eastern cities, where the integration of industries with emerging technologies has reached a certain equilibrium. In the absence of major technological innovations in the short term, the supportive role of new infrastructure in optimizing industrial structure is limited for these cities. In contrast, the industrial structure of central-western cities is still in a dynamic state of change, and with the increasing adoption of emerging technologies, the integration of industries in these cities has accelerated, due to the driving force of new infrastructure. Consequently, the marginal spillover effect of new infrastructure on industrial structure optimization is greater in central-western cities, providing more significant support for the ascent of industries towards high technology and high value-added sectors.

Furthermore, in columns (2) and (4) of Table 5, the regression coefficient of new infrastructure on the advancement of industrial structure in central-western cities is 0.0139, which is significantly greater than the regression coefficient of new infrastructure on the pace of industrial transformation, which is 0.0065. This observation aligns with the results of the baseline study, indicating that the impact of new infrastructure on the overall industrial layout and development pattern in central-western cities is greater than its effect on the pace of industrial transformation.

### 3.3.2. The Heterogeneity of Economic Scale

The heterogeneity of economic scale to some extent affects the degree of resource agglomeration, overall industrial layout, and development form of industries in cities. At the same time, the expansion of economic scale also drives the evolution of industrial structure towards rationalization and advancement. This study divides the sample into groups with smaller and larger economic scales based on per capita GDP, and verifies the heterogeneity influence of economic scale on the industrial structure optimization of new infrastructure.

**Table 5.** Heterogeneity analysis: classified by urban location.

| Variables | AIS | | SIT | |
|---|---|---|---|---|
| | (1) Eastern | (2) Central and Western | (3) Eastern | (4) Central and Western |
| NI | 0.0042 ** | 0.0139 * | 0.0018 ** | 0.0065 |
| | (0.0016) | (0.0075) | (0.0008) | (0.0042) |
| pop | −0.0027 | 0.0378 *** | −0.0020 | 0.0190 *** |
| | (0.0050) | (0.0090) | (0.0024) | (0.0042) |
| edu | 0.0049 | 0.0605 *** | 0.0031 | 0.0279 *** |
| | (0.0093) | (0.0092) | (0.0048) | (0.0043) |
| fdi | −0.1837 ** | 0.0194 | −0.0886 ** | 0.0112 |
| | (0.0811) | (0.0978) | (0.0441) | (0.0533) |
| invest | −0.0109 | −0.0017 | −0.0083 | 0.0030 |
| | (0.0132) | (0.0128) | (0.0071) | (0.0074) |
| unemp | −0.0006 | −0.0100** | 0.0003 | −0.0043 ** |
| | (0.0051) | (0.0048) | (0.0025) | (0.0021) |
| Constant | 6.6216 *** | 6.1293 *** | 2.3437 *** | 2.1043 *** |
| | (0.0327) | (0.0525) | (0.0161) | (0.0249) |
| Observations | 552 | 761 | 552 | 761 |
| Adjusted R-squared | 0.9847 | 0.9854 | 0.9785 | 0.9778 |
| year FE | YES | YES | YES | YES |
| city FE | YES | YES | YES | YES |
| F | 3.077 | 8.576 | 2.647 | 8.603 |

The standard errors are shown in parentheses. ***, **, and * represent significance at the 1%, 5%, and 10% levels, respectively.

Table 6 presents the results of the heterogeneity test, where the regression coefficient of new infrastructure is positive but not statistically significant in the group of cities with smaller economic scales, while it is significantly positive in the group of cities with larger economic scales. This indicates that new infrastructure is more beneficial for the industrial structure optimization of cities with larger economic scales, thus validating hypothesis H4. The underlying logic may be that cities with larger economic scales have advantages in information agglomeration, economic scale, complete and reasonable industrial chains, human capital, and finance, which lay the foundation for the development of new infrastructure and provide rich scenarios for the transformation and application of emerging technologies, thereby promoting the development of industries towards high-end. Conversely, for cities with smaller economic scales, their endowment of production factors, degree of industrial completeness and development level, human capital, and other advantages significantly lag behind those of cities with larger scales. On one hand, the level of development of new infrastructure is relatively low, and on the other hand, there is a lack of sufficient application scenarios, leading to insufficient optimization effect of the industrial structure of new infrastructure.

### 3.3.3. Heterogeneity in Traditional Infrastructure Construction

The new infrastructure marked by the Internet and big data, to a certain extent, relies on the development of traditional infrastructure, which serves as the foundation and support for the development of new infrastructure and empowers the expansion and extension of traditional infrastructure. The role of new infrastructure is highlighted in realizing efficiency improvement and obtaining innovative space in the traditional infrastructure industry and scene service capability, while new infrastructure empowers traditional infrastructure through the use of AI algorithm framework, 5G, and development platforms to improve facility operation and service efficiency, achieving deep integration of digital technology and industry. To explore the heterogeneous impact of traditional infrastructure on the efficiency of new infrastructure industry structure optimization, this paper takes the total paved road area at the end of the year as a proxy variable for traditional infrastructure and divides the sample into groups with relatively poor and good traditional infrastructure.

**Table 6.** Heterogeneity analysis: classified by economics scale.

| Variables | AIS | | SIT | |
| --- | --- | --- | --- | --- |
| | (1) Small Economy | (2) Large Economy | (3) Small Economy | (4) Large Economy |
| NI | 0.0095 | 0.0041 ** | 0.0053 | 0.0017 * |
| | (0.0086) | (0.0019) | (0.0055) | (0.0010) |
| pop | 0.0220 *** | 0.0070 | 0.0126 *** | 0.0024 |
| | (0.0075) | (0.0055) | (0.0041) | (0.0026) |
| edu | 0.0414 | 0.0323 *** | 0.0253 | 0.0153 *** |
| | (0.1127) | (0.0091) | (0.0542) | (0.0035) |
| fdi | −0.0678 | −0.0650 | 0.0001 | −0.0449 |
| | (0.1014) | (0.0562) | (0.0548) | (0.0374) |
| invest | −0.0115 | −0.0061 | −0.0049 | −0.0026 |
| | (0.0163) | (0.0052) | (0.0087) | (0.0026) |
| unemp | −0.0043 | −0.0045 | −0.0011 | −0.0010 |
| | (0.0059) | (0.0042) | (0.0026) | (0.0022) |
| Constant | 6.1186 *** | 6.6294 *** | 2.1117 *** | 2.3328 *** |
| | (0.0462) | (0.0352) | (0.0250) | (0.0173) |
| Observations | 709 | 675 | 709 | 675 |
| Adjusted R-squared | 0.9705 | 0.9896 | 0.9598 | 0.9860 |
| year FE | YES | YES | YES | YES |
| city FE | YES | YES | YES | YES |
| F | 1.922 | 4.306 | 1.998 | 5.310 |

The standard errors are shown in parentheses. ***, **, and * represent significance at the 1%, 5%, and 10% levels, respectively.

In groups with poor traditional infrastructure in columns (1) and (3) of Table 7, the regression coefficient of new infrastructure does not have statistical significance, indicating a strong correlation between the optimization efficiency of the industrial structure of new infrastructure and the level of traditional infrastructure. In cities with poor infrastructure, hindered by factors such as urban public service facilities, the development of new infrastructure faces greater difficulties. As a result, the flow of labor and technological factors in these areas is slow, and the allocation efficiency is lower, thereby leading to insufficient utilization of the optimization efficiency of new infrastructure's industrial structure.

**Table 7.** Heterogeneity analysis: classified by traditional infrastructure.

| Variables | AIS | | SIT | |
| --- | --- | --- | --- | --- |
| | (1) Poor Traditional Infrastructure | (2) Good Traditional Infrastructure | (3) Poor Traditional Infrastructure | (4) Good Traditional Infrastructure |
| NI | 0.0017 | 0.0053 ** | 0.0013 | 0.0024 * |
| | (0.0028) | (0.0025) | (0.0012) | (0.0012) |
| pop | 0.0039 | 0.0058 | 0.0034 | 0.0012 |
| | (0.0064) | (0.0085) | (0.0037) | (0.0046) |
| edu | 0.0863 | 0.0368 *** | 0.0550 | 0.0180 *** |
| | (0.1303) | (0.0110) | (0.0607) | (0.0050) |
| fdi | −0.1094 | −0.1435 * | −0.0448 | −0.0530 |
| | (0.1196) | (0.0739) | (0.0557) | (0.0384) |
| invest | 0.0108 | 0.0076 | 0.0058 | 0.0046 |
| | (0.0176) | (0.0203) | (0.0083) | (0.0096) |
| unemp | −0.0040 | −0.0106 ** | −0.0018 | −0.0043 ** |
| | (0.0056) | (0.0045) | (0.0027) | (0.0021) |
| Constant | 6.2764 *** | 6.5256 *** | 2.1839 *** | 2.2935 *** |
| | (0.0376) | (0.0566) | (0.0208) | (0.0305) |
| Observations | 649 | 740 | 649 | 740 |
| Adjusted R-squared | 0.9790 | 0.9843 | 0.9698 | 0.9775 |
| year FE | YES | YES | YES | YES |
| city FE | YES | YES | YES | YES |
| F | 0.470 | 3.834 | 0.693 | 3.876 |

The standard errors are shown in parentheses. ***, **, and * represent significance at the 1%, 5%, and 10% levels, respectively.

In groups with good traditional infrastructure in columns (2) and (4), the regression coefficient of new infrastructure is positive at a significance level of 10%, indicating that, in cities with good traditional infrastructure, new infrastructure can upgrade traditional infrastructure through digital technology, leading to an overall improvement in the operational efficiency, service standards, and management level of the infrastructure construction. This accelerates the aggregation of innovative resources (Shen and Shi, 2021) [49], validating hypothesis H5. At the same time, new infrastructure improves the allocation efficiency, exchange efficiency, and flow rate of production factors, enhances operational and service efficiency, and allows for a more complete release of the optimization dividend of the industrial structure of new infrastructure.

*3.4. Mediating Effects Tests*

The new infrastructure has a significant optimization effect on the industrial structure, but this conclusion only describes the overall impact between the two, and the black box of its operating mechanism needs to be opened. This article selects channels such as the "technological innovation effect" and the "industry agglomeration effect" of the new infrastructure, and uses the mediation effect model to gradually examine the mechanism path of the new infrastructure's optimization of the industrial structure.

3.4.1. Technological Innovation Effect

The output of research and development activities is reflected directly by the number of patents issued, which can also indicate the level of innovation within a city. In order to ascertain the technological innovation effects of new infrastructure construction, this study utilized the number of patent applications (gpatent) and the number of utility model patent applications (ppatent) filed in the same year as representative variables for technological innovation, and logarithmic processing was performed to test the impact of new infrastructure on technological innovation. The regression results in Table 8 show that the estimated coefficients for new infrastructure were 0.4656 and 0.3944, respectively, and were both significant at the 1% level, indicating that new infrastructure effectively promoted the level of technological innovation within cities, thereby promoting industrial upgrading. These findings are consistent with the research conclusion of He and Zhao (2021) [50]. Therefore, hypothesis H2 is verified. The improvement in new infrastructure decreases information asymmetry and transaction costs, extends the geographical scope of development, increases the accessibility of resources relied upon for innovation and the markets relied upon for scale, and enhances the transfer and flow of innovative elements and diverse information among regions. As new infrastructure construction investment increases, the spillover benefits of technological innovation are continuously released, which directly promotes the level of urban technology and provides a feasible path for promoting industrial structural optimization and upgrading.

3.4.2. Industrial Agglomeration Effect

The agglomeration model of productive service industry effectively utilizes economies of scale in the production of intermediate products and services, embedding more technology and services into the manufacturing value chain, and promoting the transformation of production processes into high value-added activities. The development of new infrastructure, based on information technologies such as artificial intelligence, cloud computing, and big data, can accelerate the integration of high value-added service industries and manufacturing industries in terms of informationization. This integration can provide a strong physical information carrier for the agglomeration of productive service industries and the synergistic agglomeration of industries, thus comprehensively enhancing total factor productivity and promoting the optimization and upgrading of industrial structure.

**Table 8.** Mechanism tests: technological innovation effect.

| Variables | gpatent | ppatent |
|---|---|---|
| | (1) | (2) |
| NI | 0.4656 *** | 0.3944 *** |
| | (0.1089) | (0.0897) |
| FDI | 22.7921 *** | 18.9695 *** |
| | (3.9600) | (3.1929) |
| fin | 0.9536 *** | 0.8946 *** |
| | (0.1487) | (0.1383) |
| invest | −0.8725 *** | −0.7957 *** |
| | (0.2430) | (0.2315) |
| urban | 0.5781 | 0.2907 |
| | (0.4057) | (0.3770) |
| expenditure | −4.5232 * | −4.3219 * |
| | (2.6786) | (2.6094) |
| Observations | 1585 | 1590 |
| Adjusted R-squared | 0.6320 | 0.6836 |
| year FE | YES | YES |
| province FE | YES | YES |
| F | 47.80 | 46.44 |

The standard errors are shown in parentheses. ***, **, and * represent significance at the 1%, 5%, and 10% levels, respectively.

For this purpose, we refer to the study conducted by Ezcurra (2006) [51] to construct the indices of professional agglomeration in productive service industries ($SP_i$) and relative agglomeration of diversity ($jac_i$), which measure the degree of specialization and diversity in productive service industry agglomerations in a certain region compared to the national level. These indices respectively reflect the level of professional agglomeration and the concentration of different productive service industries within a region. The construction method of these indices is as follows:

$$SP_i = \sum_s \left| \frac{E_{is}}{E_i} - \frac{E'_s}{E'} \right|, \tag{5}$$

$$jac_i = \frac{1}{\sum_{s=1}^{n}(E_{is} - E_s)}, \tag{6}$$

where $SP_i$ denotes the index of specialization agglomeration of the productive service industry in city $i$, $jac_i$ denotes the index of diversification of the productive service industry in city $i$, $E_{is}$ denotes the employment in the industry $s$ of the productive service industry in city $i$, $E_i$ denotes the total employment in city $i$, $E'_i$ denotes the total employment out of city $i$, $E'$ denotes the employment in industry $s$ of the productive service industry outside city $i$, and $E_s$ denotes the proportion of the productive service industry s in the national total employment. Based on Ke's (2014) [40] research and according to China's urban employment statistics, seven industries, including transportation, storage and postal services, information transmission, computer services and software, wholesale and retail, finance, leasing and business services, scientific research and technical services, environmental governance, and public facilities management are merged to represent the productive service industry.

In column (1) of Table 9, the regression coefficient of new infrastructure is 0.1463, passing the 5% significance test, which indicates that new infrastructure significantly promotes the specialized agglomeration of production-oriented service industry. The new business model—based on information technology and cloud computing platforms—empowers the agglomeration of production-oriented services and their integration into the value chain of manufacturing, breaking through the traditional industry and spatial limitations of the manufacturing industry, and increasing the technical and industrial interdependence

between the service and manufacturing industries. Therefore, new infrastructure effectively promotes the knowledge spillover and technical transfer of specialized agglomeration of production-oriented services among different industries, and more fully leverages the "optimization effect" of industrial structure, verifying hypothesis H3. The non-significant regression coefficient of new infrastructure in column (2) indicates that there is no strong correlation between new infrastructure and the diversified agglomeration patterns of production-oriented services, which may be due to the heterogeneity of the subdivided industries within the production-oriented service sector, where a high proportion of low-end production-oriented services could affect the spillover effect of new infrastructure.

**Table 9.** Mechanism tests: industrial agglomeration effect.

| Variables | SP | jac |
| --- | --- | --- |
| | (1) | (2) |
| NI | 0.1463 ** | −0.0008 |
| | (0.0709) | (0.0008) |
| FDI | −5.0517 | 0.0845 |
| | (3.0801) | (0.0668) |
| fin | −0.4036 ** | 0.0045 |
| | (0.1875) | (0.0056) |
| invest | −0.0593 | 0.0037 |
| | (0.2010) | (0.0066) |
| urban | 0.2516 | −0.0064 |
| | (0.2593) | (0.0086) |
| expenditure | 0.8302 | −0.0601 |
| | (0.8636) | (0.0771) |
| Observations | 1570 | 1568 |
| Adjusted R-squared | 0.2239 | −0.0152 |
| year FE | YES | YES |
| province FE | YES | YES |
| F | 2.525 | 0.747 |

The standard errors are shown in parentheses. ***, **, and * represent significance at the 1%, 5%, and 10% levels, respectively.

## 4. Conclusions and Discussion

Given the ongoing shift of the Chinese economy towards high-quality growth, there is an increasing recognition of the need for a reliable contemporary infrastructure system that can effectively support a modern industrial system. This study aimed to explore the correlation between new infrastructure and the optimization of industrial structure, which has received little attention in previous scholarly investigations. Consequently, our research makes an original and distinctive addition to the existing literature. This study used panel data including 266 prefecture-level cities spanning from 2011 to 2018. The results of our study indicate that the implementation of new infrastructure has a substantial impact on the advancement of industrialization at a higher level and the acceleration of industrial transformation inside urban areas. Significantly, this finding remains robust even after controlling for potential endogeneity issues. Additionally, we found regional variations in the impact of new infrastructure on industrial structure optimization, with more significant effects observed in eastern cities. This nuanced understanding of regional differences highlights our study's uniqueness. Our third key finding is that new infrastructure promotes industrial structure optimization through technological innovation and professional agglomeration effects. This provides new insights and a deeper understanding of the mechanisms behind infrastructure's impact on industrial structure, further contributing to the existing body of knowledge.

Based on these findings, this paper proposes the following policy recommendations:

Firstly, accelerate the construction of new infrastructure. Optimize the layout, structure, function, and system integration of infrastructure to better leverage the overall efficiency of the infrastructure system. Strengthen the construction of traditional infrastructure

such as railways, highways, and logistics, and promote digitalization, intelligence, and networking of traditional infrastructure. Accelerate the construction of information infrastructure, expedite breakthroughs in key common technologies of the new generation of information technology, and steadily develop integrated infrastructure. Increase investment in the construction of new infrastructure and speed up the construction of a national interconnected new infrastructure system.

Secondly, enhance the empowering level of new infrastructure for industrial development. Consolidate and strengthen the role of new infrastructure in promoting technology spillover and diffusion, and industrial technological innovation. Remove barriers and obstacles that impede the free flow of data and other production factors. Accelerate the development of the digital economy, comprehensively promote the construction of industrial Internet and the Internet of Things, empower industrial development in both supply and demand, and accelerate digital and intelligent transformation of industries. Based on digital technology such as big data, cloud computing, and artificial intelligence, deeply excavate industries and enterprises with development potential, and provide support services for their transformation and upgrading. Strengthen the innovative leadership of major scientific and technological infrastructure, build a cluster of major scientific and technological infrastructure for regional collaborative development, and provide important guarantee for industrial development.

Thirdly, promote the coordinated development of new infrastructure. Pay attention to the imbalance in the development of infrastructure between cities, accelerate the construction of new infrastructure in medium and small cities in the central and western regions, and moderately tilt policies, funds, talents, and other resources to weaker areas to eliminate the digital divide and development gap. When industrial transfer happens between cities in the east and between the east and the central and western regions, focus on supporting the transfer of new infrastructure and promote industrial localization and upgrading. Speed up the construction of comprehensive demonstration zones for the application of new infrastructure in relatively mature and developed areas to provide experience and reference for the development of local and other areas.

**Author Contributions:** Conceptualization, W.W. and Z.J.; methodology, Z.J.; software, W.W.; validation, W.W., Z.J. and H.L.; formal analysis, H.L.; investigation, W.W.; resources, Z.J.; data curation, H.L.; writing—original draft preparation, W.W.; writing—review and editing, H.L.; supervision, Z.J.; project administration, Z.J.; funding acquisition, Z.J. All authors have read and agreed to the published version of the manuscript.

**Funding:** This research was funded by Fundamental Research Funds for the Central Universities (Grant No. 2242023S30031).

**Data Availability Statement:** Publicly available datasets were analyzed in this study. The research data sources include the "China City Statistical Yearbook", EPS data platform, China Research Data Service Platform (CNRDS).

**Conflicts of Interest:** The authors declare no conflict of interest.

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
