# Peer review of "The Impact of New infrastructure Construction on Optimization and Upgrading of Industrial Structure"

_buildings, doi:10.3390/buildings13102580_

Round 1
Reviewer 1 Report
The article entitled New infrastructure and optimization and upgrading of industrial structure addresses an interesting topic.
The article contains the appropriate structure. It is correctly divided into relevant sections and their content coincides with their titles. Bibliography is correctly formulated. The scientific language of the article is mature.
My main recommendations are the following:
Abstract – The author have to clarify the aim and results of the study
Does the model you use have any limitations?
The Conclusions are vague, they should get more consistency, underlining the added value of the paper.
Author Response
Dear Reviewer,
Thank you for your thoughtful comments on our article. We appreciate your feedback and have carefully considered your suggestions. Based on your reviews and our manuscript, we have made the necessary revisions to address the issues you raised. The revised parts have been highlighted.
In response to your first concern, we have rewritten the abstract to provide a clearer and more concise explanation of the aim and results of our study as well as the limitations of the model used in this study.
In response to your comment about emphasizing the unique contributions of my paper, I have made substantial revisions to the conclusion section, which are highlighted in the revised manuscript. I now clearly state the marginal contributions that my study makes to the existing body of literature.
We sincerely appreciate your valuable feedback, as it has greatly contributed to the clarity and comprehensiveness of our article. We have incorporated your suggestions into our revisions, and we believe that the revised manuscript will address the concerns raised. Thank you once again for your input.
Best regards,
Zheng Ji

Reviewer 2 Report
Dear Authors,
The paper entitled “New infrastructure and optimization and upgrading of industrial structure.” aims to examines the impact of new infrastructure on the optimization and upgrading of the industrial structure from the perspectives of heterogeneity and impact mechanisms, and uses instrumental varia- bles to solve endogeneity problems.
After reading the article, I have the following comments and suggestions for improving it:
Abstract
I suggest improving the abstract according to the requirements of the journal, reduce unnecessary descriptions, but expand the results section.
Title
The title of the paper should be modified. It lacks reference to the research area.
Introduction
The introduction to the topic is interesting and based mainly on Chinese lierature. There are only a few world literature items out of 53. I believe that the literature review should be expanded to include world literature. There is no chapter with theoretical background in the article.
I also suggest describing in more detail why the authors believe that "building new infrastructure provides a new engine and a new impetus to optimize the industrial structure"?
Discussion chapter , the authors should discuss and explain the article's findings and results in more detail. This would contribute to significantly improving the article. The authors should compare their project and results with the results of similar studies conducted on this topic in other parts of Europe and around the world.
A Conclusions chapter is also missing.
This chapter should still answer the question: what tangible benefits this study has brought to the construction of new infrastructure and is a new impetus for optimizing the industrial structure. At the end of the article there should be recommendations.
kind regards Reviewer
Author Response
Dear Reviewer,
Thank you for your thoughtful comments on our article. We appreciate your feedback and have carefully considered your suggestions. Based on your reviews and our manuscript, we have made the necessary revisions to address the issues you raised. The revised parts have been highlighted.
We appreciate your comment on the need to broaden our literature review. While our initial focus was primarily on Chinese literature because of the specific context of our study, we acknowledge the importance of international perspectives in enriching our analysis. We agree that incorporating more world literature would provide a more comprehensive understanding of our topic. We have revisited our literature review and incorporated more references from global sources (highlighted) to provide a more balanced and representative viewpoint.
In response to your concern about the title. We agree that our current title could be more specific in this regard. To address your point, we revised the title to better represent the research area.
In response to your feedback about the abstract, we have rewritten the abstract to make it more concise and aligned with the guidelines of the journal. Unnecessary descriptions have been trimmed down to make it more succinct and focused. At the same time, I have expanded on the results section to provide a clearer and more comprehensive overview of the significant findings of our study.
In response to your comment about emphasizing the unique contributions of my paper, I have made substantial revisions to the conclusion section, which are highlighted in the revised manuscript. I now clearly state the marginal contributions that my study makes to the existing body of literature.
In response to your concern about there is no chapter of theoretical background. While we have not dedicated a separate chapter to the theoretical background, we have incorporated it into the introduction section of our article that the background lies in the concept of the digital economy and its impact on economic development. The digital economy refers to the economic activities that are based on digital technologies, such as the internet, artificial intelligence, and big data. We believe that this approach provides a smooth transition into the study's context and objectives, and ensures a comprehensive understanding of the theoretical underpinnings. However, we understand that this might not have been explicitly clear, and we appreciate your feedback. We have made the necessary revisions to make the presence of the theoretical background more apparent in our introduction as highlighted. Furthermore, we have clarified a series of recommendations based on our findings. These recommendations are designed to guide practitioners and policymakers in leveraging new infrastructure for optimizing industrial structures, stimulating economic growth, and promoting sustainable development.
We have also expanded on this in the introduction and discussion sections of the paper. We have clarified that new infrastructure, through its impact on technological innovation and professional agglomeration, can provide the necessary impetus for industrial structure optimization. Besides we have expanded our discussion chapter to provide a more detailed interpretation of our results. We have further explained how our findings contribute to the current understanding of the impact of new infrastructure on industrial structure optimization. We also We have highlighted how our findings can inform policy decisions, guide infrastructure development strategies, and contribute to improving the efficiency and effectiveness of industrial structures.
In response to your feedback about the need for a clear Conclusion chapter. We understand how important it is to have a distinct section that succinctly summarizes the key findings and implications of our study. In the original submission, the conclusion was incorporated within the Discussion section. However, we understand that this may have made it less noticeable and potentially confusing for the readers. Therefore, in response to your comment, we have revised the manuscript to rename the section “Conclusion and Discussion”.
We sincerely appreciate your valuable feedbacks, as it has greatly contributed to the clarity and comprehensiveness of our article. We have incorporated your suggestions into our revisions, and we believe that the revised manuscript will address the concerns raised. Thank you once again for your input.
Best regards,
Zheng Ji